# The Maturation Pathway of Nickel Urease

**Yap Shing Nim and Kam-Bo Wong \***

School of Life Sciences, Centre for Protein Science and Crystallography, State Key Laboratory of Agrobiotechnology, The Chinese University of Hong Kong, Hong Kong 999077, China
**\*** Correspondence: kbwong@cuhk.edu.hk

**Abstract:** Maturation of urease involves post-translational insertion of nickel ions to form an active site with a carbamylated lysine ligand and is assisted by urease accessory proteins UreD, UreE, UreF and UreG. Here, we review our current understandings on how these urease accessory proteins facilitate the urease maturation. The urease maturation pathway involves the transfer of $Ni^{2+}$ from UreE → UreG → UreF/UreD → urease. To avoid the release of the toxic metal to the cytoplasm, $Ni^{2+}$ is transferred from one urease accessory protein to another through specific protein–protein interactions. One central theme depicts the role of guanosine triphosphate (GTP) binding/hydrolysis in regulating the binding/release of nickel ions and the formation of the protein complexes. The urease and [NiFe]-hydrogenase maturation pathways cross-talk with each other as UreE receives $Ni^{2+}$ from hydrogenase maturation factor HypA. Finally, the druggability of the urease maturation pathway is reviewed.

**Keywords:** urease maturation; metallochaperone; nickel; G-protein; conformational change

---

## 1. Introduction

Urease catalyzes the hydrolysis of urea into carbon dioxide and carbamate, which spontaneously decomposes into ammonia and another carbon dioxide. The enzyme is involved in nitrogen metabolism that is found in bacteria, archaea, fungi, plants, and some invertebrates [1–3]. While the Jack bean (*Canavanlia ensiformis*) urease was the first nickel enzyme identified [4] and the first enzyme to be crystallized [5], the mechanism of urease maturation is the most well studied in bacteria, in particular *Klebsiella aerogenes* and *Helicobacter pylori*. Urease has been implicated in the pathogenesis of bacterial infections. For example, *H. pylori* can colonize in the acidic stomach due to the ureolytic activity of urease [6]. *H. pylori* infection increases the risk of peptic ulcer and gastric cancer [7,8]. Most of the ureases discovered so far are nickel enzymes. One notable exception is an iron urease found in *Helicobacter mustelae* in ferret stomach [9].

The concentration of free nickel ions is tightly regulated in cells because $Ni^{2+}$ can inactivate enzymes by displacing weaker ions such as $Mg^{2+}$ in the active sites [10,11]. To avoid cytotoxicity, cells have to evolve a mechanism to deliver nickel from one protein to another without releasing the toxic metal to the cytoplasm. In the urease maturation pathway, there are four urease accessory proteins, UreD, UreE, UreF and UreG, involved in the nickel delivery. In this article, we review how these metallochaperones interact with each other and with urease to facilitate the transfer of nickel ions in the urease maturation pathway. Correct metalation is ensured by specific protein–protein interactions that are allosterically regulated by binding/hydrolysis of guanosine triphosphate (GTP). We also review the cross-talking between the maturation pathways of ureases and [NiFe]-hydrogenases, urease maturation in plants and the potential of the urease maturation pathway in antibacterial drug discovery. Finally, some unanswered questions on the urease maturation pathway are discussed.

## 2. Structures of Urease

The urease sequence is highly conserved. Most of the bacterial ureases are comprised of three subunits [12]. For example, *K. aerogenes* urease has three subunits: α-subunit (UreC), β-subunit (UreB) and γ-subunit (UreA) (Figure 1A) [13]. In *H. pylori*, the two smaller β and γ subunits are fused to form the UreA, and the large α-subunit is renamed as UreB [14]. In plants, all three subunits are fused into one polypeptide chain as UreA [15]. In *K. aerogenes* [16,17] and *Sporosarcina pasteurii* [18], the urease contains three catalytic sites constituted by the α, β and γ subunits. Three copies of the subunits UreABC form a trimeric disc-like structure (Figure 1A). In plant urease, the disc-like structure dimerizes to form a hexameric urease (UreA$_3$)$_2$ (Figure 1B) [19]. *H. pylori* UreA contains a 27-residue C-terminal extension that is responsible for the formation of a dodecameric quaternary structure (UreAB$_3$)$_4$ [20].

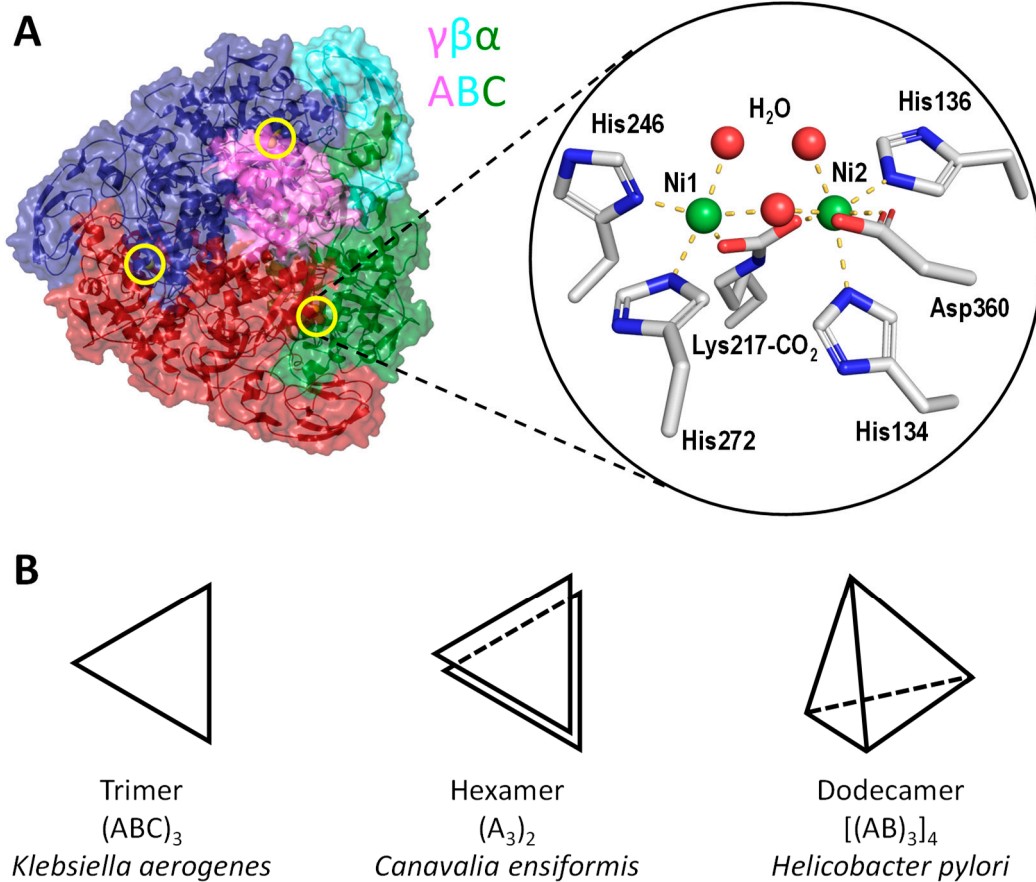

**Figure 1.** Structures of ureases. (**A**) Crystal structure of *K. aerogenes* urease (PDB: 1FWJ [16]). The basic catalytic unit of ureases consists of α, β and γ subunits encoding by *ureC*, *ureB* and *ureA*, respectively. In *K. aerogenes* and *S. pasteurii*, three copies of UreABC form a trimeric disk-like structure. The positions of three actives sites are circled in yellow and zoomed in at the right. The two nickel ions are coordinated by a carbamylated lysine residue, four histidine, one aspartate, and three water molecules. (**B**) The trimeric urease, schematically represented as triangles, forms the basic repeating unit of ureases to form more complex quaternary structures. In plant ureases, such as *Canavalia ensiformis* (PDB: 3LA4 [19]), two urease trimers are stacked together in a C2 symmetry to form a hexameric quaternary structure. In *H. pylori*, four trimers assemble in a tetrahedral symmetry to form a dodecameric urease (PDB: 1E9Z [20]).

The urease active site is located in the α-subunit and is highly conserved. It contains two Ni$^{2+}$ ions [16,18–20] bridged by the carboxyl group of a carbamylated lysine [17,20,21], and chelated by four histidine residues and one aspartate residue (Figure 1A). Purified *K. aerogenes* urease can be activated in vitro by incubation with high concentration of bicarbonate and Ni$^{2+}$ [22]. The carbamylation is

likely to come from carbon dioxide, instead of bicarbonate as suggested by the result of pH jump experiment [22]. Mutagenesis studies suggest that the carbamylated lysine residues are essential to urease maturation [23]. Urease activity can be inhibited by $Zn^{2+}$, $Cu^{2+}$, $Co^{2+}$ and $Mn^{2+}$ [24].

## 3. Genetic Studies Showed the Importance of Urease Accessory Proteins

In *K. aerogenes*, the urease operon is arranged as *ureDABCEFG* [25,26] (Figure 2). *ureA*, *ureB* and *ureC* respectively encode the γ, β and α subunits of the urease structural genes. *ureD*, *ureE*, *ureF* and *ureG* encode for the urease accessory proteins essential for the maturation of urease. In *H. pylori*, the operon is arranged as *ureABIEFGH* [27,28]. There is a unique *ureI* encoding for an acid-gated urea channel for colonization in the acidic stomach [29,30]. *ureH* is an orthologue of *ureD*. For simplicity, UreD is used in this article to denote the protein product of *ureH* in *H. pylori* or *ureD* in other species.

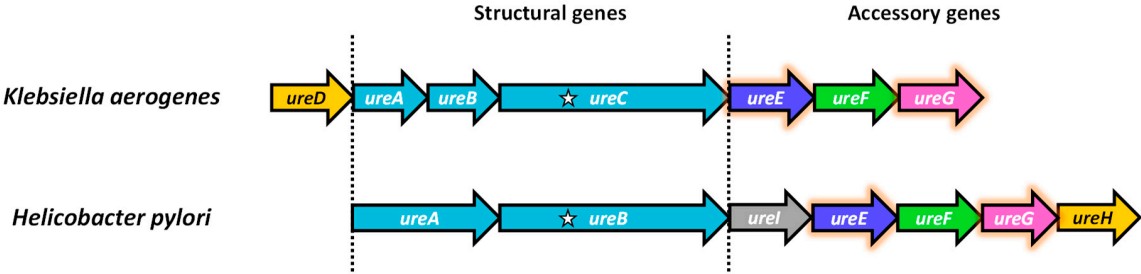

**Figure 2.** Comparison of *K. aerogenes* and *H. pylori* urease operons. Orthologous genes are indicated using the same color. The locations of the active-site lysine residue involved in carbamylation and nickel binding are indicated as stars. *H. pylori* contains an extra *ureI* gene encoding for an acid-gated urea channel.

Bacterial cells harboring the urease operon showed nickel-dependent activation of urease [26,31,32]. Urease purified from cells containing the intact urease operon was active, whereas the urease purified from cells containing only the urease structural genes was inactive [33,34]. This observation suggests that the urease maturation is assisted by the urease accessory proteins. Deletion [26], knockout [35] and transposon mutagenesis [28] of individual accessory genes, *ureG*, *ureF* and *ureD*, abolished urease activity. The activity could be partially [26] or fully [32] recovered by gene complementation. Deletion of *ureE* either lowered [26] or abolished the urease activity [32,35,36], which was partially regenerated by adding nickel [32]. These results suggested that UreG, UreF and UreD are absolutely required for urease maturation, while UreE facilitates the process.

## 4. The Formation of UreGFD Complex

Urease apoprotein was shown to interact with UreD [37,38], UreF/UreD [39,40], or UreG/UreF/UreD [38,40–42]. It has been known that UreD and UreF can form a UreFD complex, which can then recruit UreG to form a UreGFD complex [43]. The UreGFD complex can form an activation complex [43,44] with urease apoprotein and activate urease in a GTP-dependent manner [43,44]. Substitution of the P-loop residues of UreG abolished its ability to activate urease [45,46].

The structural studies provided insights into how UreD, UreF and UreG interact with each other to form the protein complexes required for urease maturation. UreD forms a complex with UreF [35,38,40,47]. Crystal structure of the *H. pylori* UreFD complex revealed that it is a 2:2 heterodimer with UreF at the middle providing the dimerization interface (Figure 3A) [48]. UreD bound to both ends of the UreF dimer, forming a rod-shape head-to-head dimer of heterodimer UreFD (Figure 3A). UreF contains highly conserved residues at its C-terminal tail, which is unstructured and susceptible to proteolytic cleavage when expressed alone [48,49]. The C-terminal tail of UreF was shown to be essential for the interaction with UreD [40,48], the assembly of an activation complex [40], and urease activation [40,48]. Upon binding to UreD, these C-terminal residues become structured and form

an extra helix-10 and a loop structure stabilized by hydrogen bonds involving a conserved Arg-250 residue [48]. These conformational changes were shown to be important for recruiting UreG to form the UreGFD complex by mutagenesis studies. For example, the R250A variant abolished the formation of UreGFD complex and urease activation [43].

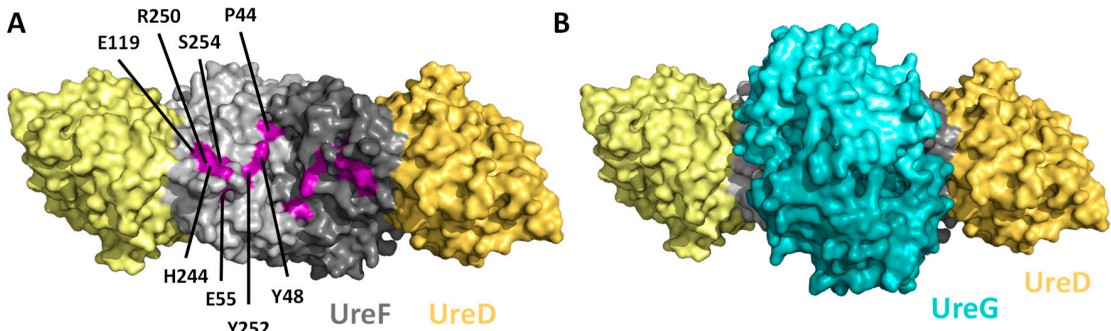

**Figure 3.** Dimerization of UreF creates a saddle-like structure for recruiting UreG to the UreGFD complex. **(A)** Crystal structure of the UreFD complex (PDB: 3SF5 [48]). UreF (grey) interacts with UreD (yellow) to form a 2:2 heterodimer in a rod-shape topological arrangement of D:F:F:D. A saddle-like structure is formed at the UreF dimer surface. Invariant residues of UreF that have been shown to be essential to UreG binding and urease activation are highlighted in purple and are numbered according to the *H. pylori* sequence [48,50]. **(B)** Crystal structure of the UreGFD complex showing that UreG dimer (cyan) binds to the saddle-like structure of UreF (PDB: 4HI0 [43]).

The crystal structure of the *H. pylori* UreGFD complex (PDB: 4HI0) revealed that it is a dimer of the heterotrimers of UreGFD [43] (Figure 3B). The complex contains a guanosine diphosphate (GDP)-bound UreG dimer sitting on the UreF wherein the dimerization axes of UreG and UreFD are almost perpendicular to each other. UreF-mediated dimerization is required to provide a complete UreG binding site. Substitutions that broke the dimerization of UreF abolished the recruitment of UreG to the UreGFD complex and urease maturation [40,43]. UreG binds to a saddle-like structure of UreF consisting of clusters of conserved residues [43]—some of them have been identified to be important for recruiting UreG to the activation complex and for activating urease [50] (Figure 3A).

## 5. UreG Dissociates from the UreGFD Complex and Forms a Dimer in the Presence of Ni/GTP

That UreG undergoes Ni/GTP-dependent dimerization was identified when $Ni^{2+}$ and GTP were added to the GDP-bound UreGFD complex, which was then dissociated into a UreG dimer and the UreFD complex [43]. UreG is a nickel chaperone [43,51,52] and a SIMIBI (after Signal recognition particle, MinD and BioD) class GTPase [53]. UreG remains as a monomer in the absence of GTP and it binds $Ni^{2+}$ with lower affinity [54,55]. UreG only dimerizes when both GTP and $Ni^{2+}$ were present, and the UreG dimer binds one $Ni^{2+}$ per dimer with a $K_d$ of 0.36 μM [43,52]. $Zn^{2+}$ can also induce dimerization of UreG [54,56,57], but the Zn/UreG dimer was not stable and dissociated to monomer when the excess $Zn^{2+}$ was removed by gel filtration [43]. Moreover, this Zn/UreG dimer is inactive in GTP hydrolysis [52]. Dimerization is required for GTP hydrolysis. The cysteine and histidine in the conserved CPH motif (Cys-Pro-His) are important for UreG dimerization and nickel binding [43,54], as well as urease activation [43,46,58]. After GTP hydrolysis, the UreG dimer dissociates back to monomer and releases one $Ni^{2+}$, providing a plausible mechanism for coupling GTP hydrolysis to nickel delivery [43]. GTP-dependent dimerization [59–63] and conformational changes [64] were also observed in HypB, another SIMIBI GTPase involved in the [NiFe]-hydrogenase maturation pathway.

Structural insights into how $Ni^{2+}$ and GTP induce dimerization of UreG were provided by the crystal structure of the UreG dimer in complex with $Ni^{2+}$ and GMPPNP, a nonhydrolyzable analogue of GTP [51]. The GTP binding pocket is sandwiched between two UreG chains and a $Ni^{2+}$ ion is coordinated by the conserved CPH motif from each chain [51]. The structure of Ni/GMPPNP-bound

UreG dimer is compared to that of the GDP-bound UreG in the UreGFD complex (Figure 4). Upon GTP binding, the γ-phosphate of GTP introduces a charge–charge repulsion on Asp37 in the G2 switch, initiating a swinging motion of helix-2, and Glu42 forms a hydrogen bond with Arg130 of the opposite chain. Consequently, the zip-up motion of β2 and β3 propagates the conformational changes to the CPH motif, where Cys66 and His68 reorientate towards the dimeric interface to chelate a $Ni^{2+}$ ion in a square-planar geometry (Figure 4B) [51]. The structural changes observed also explain why UreG dissociates from the UreGFD complex upon GTP binding as residue Tyr39 in the G2 region swings outward and makes steric clashes with UreF [48].

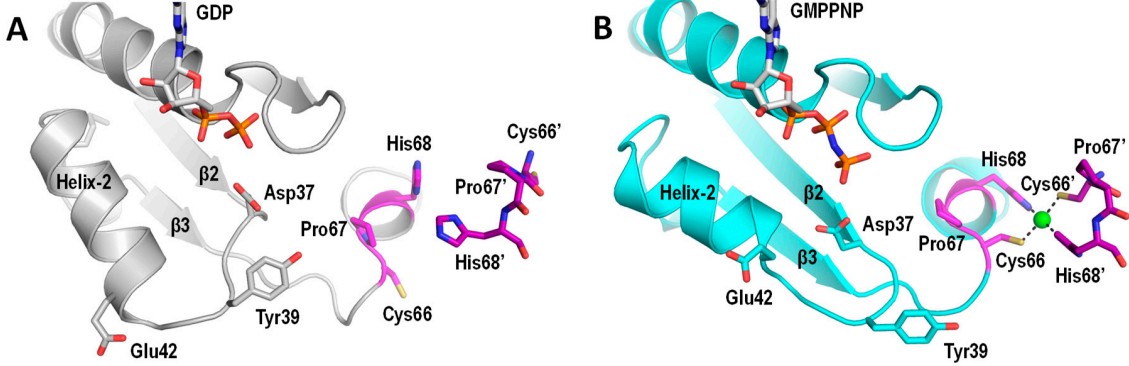

**Figure 4.** Guanosine triphosphate (GTP)-dependent conformational changes of UreG. (**A**) The structures of GDP-bound *H. pylori* UreG (PDB: 4HI0 [43]) is compared to the (**B**) Ni/GMPPNP-bound *K. pneumoniae* UreG (PDB: 5XKT [51]). Invariant residues involved in the conformational changes and nickel binding are indicated and numbered according to the *H. pylori* sequence. Residues in the opposite chain are numbered with apostrophes. Noteworthy, Cys66 and His68 of the CPH motif (magenta) are pointing away from each other in the GDP-bound UreG. Upon binding of GMPPNP, they reorient to form a square-planar nickel binding site at the dimeric interface. Moreover, the swinging motion of Tyr39 creates steric clashes that induce dissociation of UreG from the UreGFD complex.

## 6. UreG–UreE Interaction Is GTP-Dependent

UreG interacts with UreE in 2:2 ratio to form a $UreE_2G_2$ complex in the presence of GTP and $Mg^{2+}$. The formation of $UreE_2G_2$ is independent of $Zn^{2+}$ and $Ni^{2+}$ [52]. In the presence of GDP, the complex dissociates into the $UreE_2G$ complex and a UreG monomer [52]. Interestingly, addition of $Ni^{2+}$ and GTPγS to the UreGFD complex and UreE promotes UreG to switch its protein binding partner from UreFD to UreE, resulting in the formation of the $UreE_2G_2$ and UreFD complexes [51]. The D37A/E42A variant of UreG failed to dissociate from the UreGFD complex and formed the Ni/GTP-bound UreG dimer, presumably disrupting the conformational changes induced by GTP binding [51]. Interestingly, the D37A/E42A variant also failed to form the $UreE_2G_2$ complex in the presence of Ni/GTP, further supporting that the conformational changes in UreG are important for the UreE–UreG interaction [51]. The formation of the $UreE_2G_2$ complex facilitates the transfer of $Ni^{2+}$ from UreE to UreG. By monitoring the thiolate-to-$Ni^{2+}$ transition at 337 nm, it has been shown that $Ni^{2+}$ was transferred from UreE to UreG within the $UreE_2G_2$ complex in its GTP-bound state, but not in the $UreE_2G$ complex in the presence of GDP [52].

How UreE interacts with UreG is not known. Crystal structures of UreE from *S. pasteurii* [65], *K. aerogenes* [66] and *H. pylori* [67,68] have been solved, and share high structural homology. UreE exists as a dimer in solution [68–71]. High protein concentration and the presence of $Zn^{2+}$, $Cu^{2+}$ or $Ni^{2+}$ can induce UreE tetramerization [65,72–74], but mutagenesis studies showed that the formation of the tetramer was not essential to urease activation [68]. The C-terminal domain of two UreE chains interact with each other to form a dimer. At the dimeric interface, UreE has one conserved metal-binding site formed by His102 (numbered according to the *H. pylori* sequence) in a GNXH motif from each of the UreE chain [65–68]. This central histidine is essential to urease maturation [68,75,76].

The variable C-terminus histidine-rich tail of UreE [76] has been shown to bind $Ni^{2+}$ (residue 143–157 in *K. aerogenes* [70,72,77]; residue 137–147 in *S. pasteurii* [73]). In the crystal structure of *H. pylori* UreE, an extra histidine (His152) of this C-terminal tail was shown to bind a $Ni^{2+}$ or $Zn^{2+}$ at the dimeric interface (Figure 5B) [67,68]. Neither the H102A substitution nor truncation of the variable C-terminal tail (residue 158–170 of *H. pylori* UreE) affect the formation of the $UreE_2G_2$ complex [52]. Truncation of the C-terminal tail of *K. aerogenes* UreE showed a 25–60% decrease in urease activation [77,78]. The functional role of the C-terminal tail is unclear. However, it has been proposed that the C-terminal tail may play a role in regulating the binding and release of $Ni^{2+}$ [65,67]. Based on charge and shape complementarity, models of how UreE interacts with UreG have been proposed [71,79,80]. The models predict that the nickel binding sites of UreE and UreG should point towards each other, which is supported by the mutagenesis studies that show R101A UreE or C66A UreG destabilized the $UreE_2G_2$ complex [52].

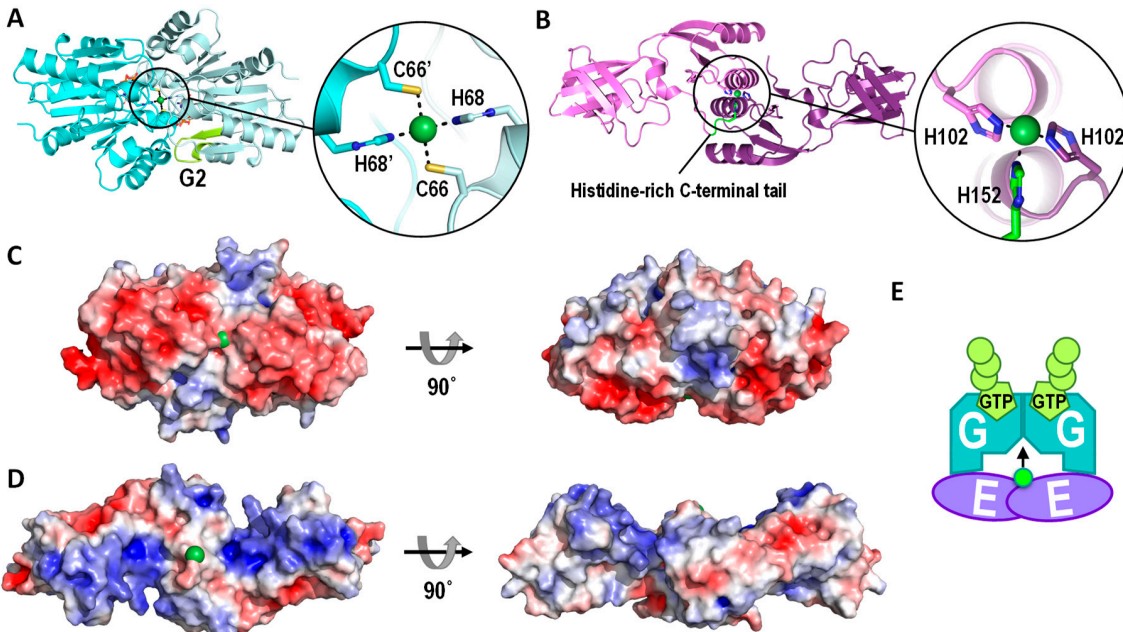

**Figure 5.** Structures of UreE and UreG. Cartoon representation of *K. pneumoniae* Ni/GMPPNP-bound UreG (**A**, PDB: 5XKT [51]) and *H. pylori* Ni-bound UreE (**B**, PDB: 3TJ8 [67]). Both UreG and UreE bind a nickel ion at the dimeric interface. The surface electrostatic potentials of (**C**) UreG and (**D**) UreE were calculated using the APBS program [81] and are color coded (red, $-5$ kT/$e$; blue, $+5$ kT/$e$). (**E**) It has been suggested that UreG and UreE are complementary in charge and shape and are likely to form a $UreE_2G_2$ complex with their nickel binding sites pointing towards each other [71,80].

## 7. How Urease Accessory Proteins Facilitate Urease Maturation

GTP-dependent conformational changes of UreG provide a mechanism where GTP binding/hydrolysis facilitates the delivery of nickel along the urease maturation pathway (Figure 6). It has been shown that the Ni/UreE dimer, providing the sole nickel source, can activate urease in the presence of UreGFD and GTP [51]. Binding of GTP induces UreG to dissociate from the UreGFD complex and bind with UreE to form the $UreE_2G_2$ complex. The $UreE_2G_2$ complex, which can also activate urease in the presence of UreFD complex [51], facilitates the transfer of nickel from UreE to UreG [52]. Direct protein–protein interactions among the urease accessory proteins are required as separating Ni/UreE and UreGFD/urease by a dialysis membrane abolished the urease activation in vitro [51]. After UreG gets its $Ni^{2+}$, it can interact with UreFD and urease apoprotein to form the activation complex [43]. Mutagenesis studies suggested that UreG binds to UreFD in the activation complex [46,50]. That Ni/GTP-bound UreG, which dissociates from the UreFD complex, can interact

with UreFD in the activation complex suggests that UreFD may undergo conformational changes upon binding of urease apoprotein to accommodate Ni/GTP-bound UreG in the activation complex. Whether UreE is involved in this activation complex is unclear. It has been shown that UreG can pull down UreE, UreD, UreF and urease [46,82]. Given that UreG can also interact with UreE, this observation did not conclusively demonstrate the interaction of UreE to the activation complex. Furthermore, it has been shown that urease can be activated in the presence of Ni/UreG, UreFD and urease apoprotein without UreE [51], suggesting that UreE is not essential in the formation of the activation complex.

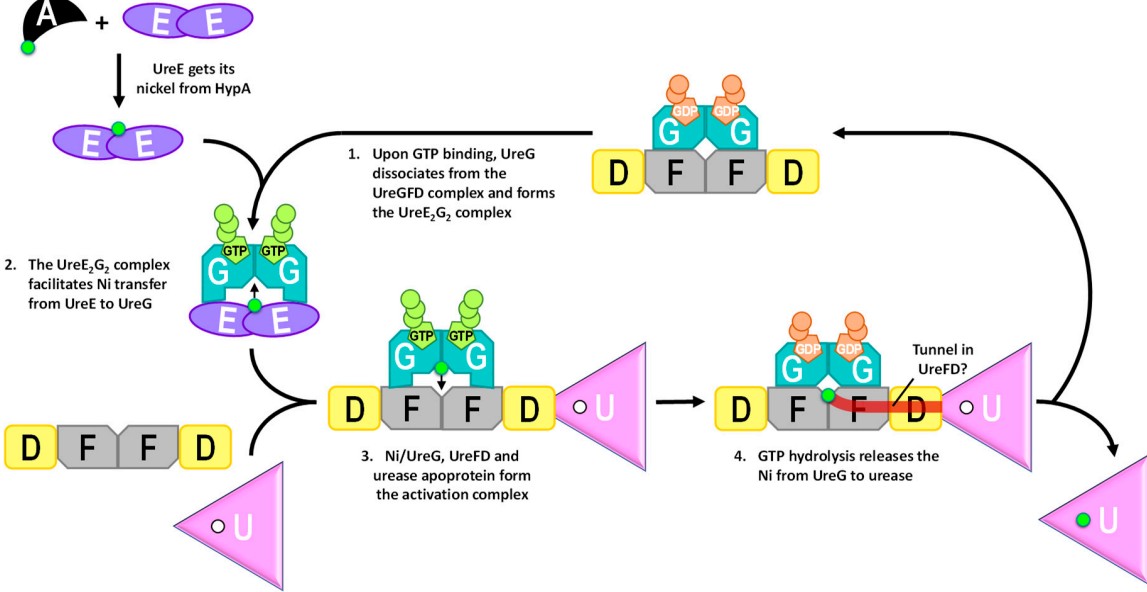

**Figure 6.** The urease maturation pathway.

It is well established that GTP hydrolysis is essential to urease maturation [41,83]. Substituting the conserved Lys-20 and Thr-21 residues to alanine in the P-loop region of *K. aerogenes* UreG abolished its ability to bind ATP-linked agarose, the formation of the activation complex and urease maturation [45]. As the Ni/UreG dimer will dissociate into monomers and release the bound $Ni^{2+}$ upon GTP hydrolysis [43], the GTPase activity of UreG must be tightly regulated to prevent nonproductive release of $Ni^{2+}$ outside the activation complex. It has been shown that the GTPase activity is only detectable by addition of bicarbonate [43]. One hypothesis is that bicarbonate can serve as one of the substrates of UreG that results in the formation of an intermediate of a carboxy-phosphate [41]. The effect of bicarbonate on the UreG activity exhibited a classical Michaelis–Menten kinetics ([52], Figure 7). Using a saturated concentration of GTP, we estimated the $K_m$ value of bicarbonate was 13 ± 4 mM. In addition to bicarbonate, GTPase activity of UreG can also be increased by ~2–3 fold by addition of UreE, $NH_4^+$ or $K^+$ [52]. Even under the most favorable conditions, the activity of UreG is still very low. In our hands, the turnover number ($k_{cat}$) of *H. pylori* UreG is ~0.006 $s^{-1}$ at 37 °C in the presence of $K^+$ and saturated concentration of GTP (Figure 7). The low intrinsic activity of UreG should be of advantage as it prevents premature hydrolysis that releases the bound $Ni^{2+}$ to the cytoplasm. It is unclear how and when the GTPase activity of UreG is activated. One possibility is that the GTPase activity of UreG is stimulated by the conformational changes induced during the formation of the activation complex. Conformational changes in the formation of the UreFD/urease complex have been suggested by cross-linking experiments [84].

GTP hydrolysis changes the conformation of the CPH motif in such a way that the square-planer coordination by Cys66 and His68 is disrupted (Figure 4), promoting the release of $Ni^{2+}$ in the activation complex. Interestingly, it has been shown that interaction between UreG and UreF is essential to the urease maturation [46,48,50]. Mutations that broke the UreG–UreF interaction also abolished urease maturation, suggesting that UreG is likely bound to the activation complex via the UreFD complex.

As the nickel binding site of UreG is far away from the active site of urease, it is not fully understood how the $Ni^{2+}$ is transferred from UreG to urease after GTP hydrolysis. One intriguing hypothesis suggested that the $Ni^{2+}$ reaches urease through a tunnel within the core of the UreFD complex [82,85,86]. Large cavities have been identified in the structures of *H. pylori* UreFD complex [82,85,86]. The cavities form a tunnel, which is wide enough for $Ni^{2+}$ to pass through, connecting the nickel binding site of UreG to an exit at UreD. The tunnel hypothesis is supported by: (1) Both UreF and UreD were able to bind $Ni^{2+}$ [38,86]; (2) Tunnel-disrupting variants of UreD were shown to greatly reduce the urease maturation without affecting the UreD–urease interaction or the formation of the UreGFD complex [82,86]. As UreD is responsible for binding urease in the activation complex, the tunnel within the UreFD complex should provide a mechanism for the transfer of $Ni^{2+}$ from UreG to the urease within the activation complex.

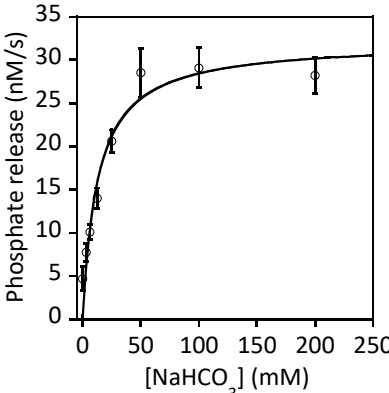

**Figure 7.** Dependency of GTPase activity of UreG on bicarbonate concentration. GTPase activity of *H. pylori* UreG (5 μM) was measured at 37 °C in 0–200 mM $NaHCO_3$ at saturated concentration of GTP (2 mM) in 200 mM KCl, 2.5 μM $NiSO_4$, 2 mM $MgSO_4$, 1 mM TCEP, 20 mM HEPES pH 7.5 by following the concentration of phosphate released as described previously [87]. Fitting the data to the Michaelis–Mention equation yielded a $K_m$ value of 13 ± 4 mM and a $k_{cat}$ value of 6.4 ± 0.6 × 10$^{-3}$ s$^{-1}$.

## 8. UreE Gets Its Nickel from Cross-Talking to the Hydrogenase Maturation Pathway

Hydrogenase maturation factors HypA and HypB, which are responsible for the delivery of nickel ions to the active site of [NiFe]-hydrogenase [88–94], are also essential for urease maturation [31,90]. Genes of hydrogenase maturation factors were first identified in *Escherichia coli* [94,95], where *hypA* and *hypB* are encoded in the *hyp* operon. In *H. pylori*, the two genes are located in different locations in the chromosome [96]. Urease activity was greatly reduced in *hypA* or *hypB* knockout strains of *H. pylori* [31,90]. The urease activity could be restored by addition of $Ni^{2+}$ in the growth medium. These observations suggest that the hydrogenase maturation factors HypA and HypB are required for the urease maturation. Structures of HypA [97,98] revealed a mixed α/β topology containing a nickel binding domain with micromolar affinity [99–102], and a zinc binding domain [98,100,102–105] with nanomolar affinity [102]. Residues coordinating the $Zn^{2+}$ [100,105] and $Ni^{2+}$ [36,99,101] are essential for urease maturation. HypA interacts with the C-terminal domain of UreE [52,106–109] to form a 1:2 complex [52,101,107,108]. The complex formation creates a unique nanomolar affinity nickel-binding site, which is not found in any of the individual proteins [107]. $Ni^{2+}$ can be transferred from HypA to UreE in the complex [52,108]. It has been shown that HypA and UreG compete with each other for UreE [109]. In the presence of $Mg^{2+}$ and GTP, UreE dissociates from the HypA/$UreE_2$ complex and interacts with UreG to form the $UreE_2G_2$ complex, suggesting the switching of the protein binding partners is GTP-dependent [52]. It has been demonstrated that $Ni^{2+}$ can be transferred from HypA to UreE to UreG [52]. HypA, on the other hand, obtains its $Ni^{2+}$ from HypB [60,110] with the help of SlyD [106,111–114] of the hydrogenase maturation pathway, which has been reviewed in [115,116].

## 9. Urease Maturation Pathway in Plants

In plants, urease and its accessory genes contain introns and are scattered on different chromosomes. The urease structural genes and homologues of UreD, UreF and UreG have been identified in plants [117,118]. Similar to the maturation of bacterial ureases, the formation of the UreGFD/urease activation complexes was shown to be essential to the maturation of ureases in rice (*Oryza sativa*) [119,120] and *Arabidopsis* [120,121]. Plant UreG, previously named as p32, is encoded by *Eu3* [122]. It has a GTPase domain homologous to bacterial UreG and an histidine-rich N-terminal extension and a plant-specific region containing two HXH motifs [120]. The two HXH motifs, but not the histidine-rich extension, were found to be essential for urease activation [120]. Interestingly, no UreE homologue was found in plants. It was postulated that the plant UreG, with the plant specific HXH motifs, may combine the function of bacterial UreE and UreG into one protein [56,120–122].

## 10. Urease Maturation Pathway Is Druggable

Urease is a microbial virulence factor for gastric infection by *H. pylori*, urinary tract infection by *Proteus mirabilis* and *Klebsiella* strains, and lung infection by fungus *Cryptococcus neoformans* [123]. The first line treatment of *H. pylori* infection is combining proton-pump inhibitors with antibiotics in the triple therapy [124]. However, the emerging antibiotic resistance make the development of new drugs pressing. Recent research findings suggest that the urease maturation pathway could become a new druggable target [65,123]. Bismuth is a heavy metal that is effective for *H. pylori* eradication [124]. Recent study showed that colloidal bismuth subcitrate inhibits the biosynthesis of active *H. pylori* urease by displacing $Ni^{2+}$ in UreG [125]. Virtual screening has identified two compounds that inhibit the GTPase activity of UreG, reduce urease activity with higher potency than clinically using acetohydroxamic acid, and suppress *H. pylori* infection in a mammalian cell model [125]. These results demonstrate that the urease maturation pathway is a novel druggable target for treatment of *H. pylori* infection.

## 11. Future Perspectives

While major progress has been made on how urease accessory proteins interplay to deliver $Ni^{2+}$ to ureases, some interesting questions remain unanswered. (1) The structure of the activation complex is not known despite several structural models that have been proposed previously [126,127]. A high-resolution structure of the activation complex should provide novel structural insights into how GTPase activity of UreG is stimulated and the mechanism of nickel transfer within the complex. (2) The mechanism of carbamylation of the active site lysine is relatively understudied. Does it take place before or after the insertion of $Ni^{2+}$? Is the carbamylation reaction catalyzed by other factors? (3) Urease requires two $Ni^{2+}$ to function. Since only one $Ni^{2+}$ ion is carried per UreG dimer, it takes two round trips of UreG to activate urease. MBP-UreD and MBP-UreFD were shown to dissociate from urease after activation [37,39]. It is unclear what triggers the dissociation of the activation complex after urease activation. (4) What could be the driving force for the transfer of $Ni^{2+}$ from UreG to the urease through the tunnel? How is the free energy change of GTP hydrolysis coupled to the nickel transfer? (5) Unlike the release of $Ni^{2+}$ from UreG that can be explained by GTP hydrolysis, it is not known why nickel is transferred unidirectionally from HypA to UreE and from UreE to UreG. Structure determination of the $HypA/UreE_2$ and $UreE_2G_2$ complexes should help to provide structural insights to this question.

## 12. Conclusions

The urease maturation pathway represents the most well-studied paradigm on how nickel ions are delivered from one protein to another through specific protein–protein interactions. One central theme depicts the binding/hydrolysis of GTP allosterically regulate the switching of protein-binding partners and the transfer of nickel ions from UreE → UreG → UreF/UreD → urease. This mechanism ensures

that the nickel ions are always protein-bound to avoid leaking of the toxic metal to the cytoplasm. UreE also receives its nickel ions from interacting with HypA of the hydrogenase maturation pathway. A better understanding of the urease maturation pathway allow us to develop drugs against the *H. pylori* infection.

**Funding:** This work was funded by the grants from the Research Grants Council of Hong Kong (14117314, AoE/M-05/12, and AoE/M-403/16) and from the Research Committee of The Chinese University of Hong Kong (3132814 and 3132815).

**Conflicts of Interest:** The authors declare no conflicts of interest.

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
