# Peer review of "The Maturation Pathway of Nickel Urease"

_inorganics, doi:10.3390/inorganics7070085_

Round 1
Reviewer 1 Report
The authors have provided a succinct and easy-to-read review that covers previously published studies on the maturation of urease. The issues, indicated by line number in the PDF, are rather minor:
10: The phrasing needs to be improved for “insertion of nickel ions to a carbamylated lysine at the active site” such as “insertion of nickel ions to form an active site with a carbamylated lysine ligand”.
16: change “nickels” to “nickel ions”.
17: insert “the” before “protein”.
27: insert “the” before “mechanism”.
35: change “delivery” to “deliver”.
38: change “interacts” to “interact”.
45: Start the sentence with “The”.
50: fix “consituted”.
62: The figure incorrectly shows UreA as cyan and UreB as magenta; these are reversed. Also the enlarged view incorrectly shows His219 as a ligand; it is 3.6 Å from the metal and not bound.
64: change “is consisted” to “consists”.
67: The text is incorrect both because only one Ni has octahedral coordination and because only 4 histidines are ligands.
76: Perhaps mention the locations of hypA and hypB?
77: insert “the” before “acidic”.
86: insert “results” before “suggested”.
94: “Apo-urease” was not defined, so consider replacing with “Urease apoprotein”.
97: same as above. Also, mutations occur in DNA, not proteins. Thus, change “Mutation of the P-loop” to “Mutation of the bases encoding the P-loop” or change “Mutation” to “Substitution”.
101: Insert “The” to start the sentence.
105: move “segment” earlier to after “tail” or delete.
110: A mutant is a cell containing a mutation, not an altered protein. Change “mutant” to “variant” or “mutein”.
112: Insert “The” to start the sentence and “the” before “heterotrimers”.
115: Change “Mutations” to “Substitutions”.
117: Change “consisted” to “consisting”.
126: Change “shows” to “showing”.
136: Italicize the thermodynamic constant.
135: insert “the” before “Zn”.
137: change “GTPase hydrolysis” to “GTP hydrolysis”.
139: insert “the” before “UreG”.
143: change “was” to “were”.
144: insert “the” before “UreG”.
145: A protomer is the smallest unit composed of at least two different chains that form a larger oligomer; thus, the word is used improperly here (change to “chain”).
146: same comment.
148: insert “the” before “G2”.
150: fix “protomer”.
154: change “swing” to “swings” and “making” to “makes” (or add comma and delete “and”).
159: “promoter” is wrong in two ways; change to “chain”.
161: change “orientate” to “reorient”.
166: delete “the” before “UreE2G2”.
167: change “dissociated” to “dissociates” and add “the” after “into”.
168: insert “its” after “switch”.
170: change “mutant” to “variant”.
172: same comment.
173: change “is” to “are”.
179: change “shared” to “share”.
183: change “protomer”.
185: same comment.
186: change “have” to “has”.
188: change “binding” to “bind”.
189: change “mutation”.
195: change “predicted” to “predict”.
196: change to “studies that show R101A UreE or C66A UreG destabilized”.
199: Consider revising panel E so the GTP molecules exhibit a more realistic position instead of being on the periphery of the protein as shown.
215: change to “urease apoprotein”.
219: change “Nevertheless” to “Furthermore”
220: change to “urease apoprotein”.
222: change “Mutating” to “Substituting”.
224: change “binding” to “bind”.
230: change “resembled a” to “exhibited”.
231: insert “a” after “Using” and use italics for the kinetic parameter.
232: change “folds” to “fold”.
234: change “hand” to “hands” and use italics for the kinetic constant.
242: fix “planner”.
244: fix “Mutations”.
247: change to “site of UreG is far”.
248: change “from the UreG to the urease” to “from UreG to urease”.
253: fix “mutations”.
262: change “phosphates” to “phosphate”.
263: italicize two kinetic constants.
268: change “with” to “containing”.
280: The GTP and GDP molecules should be located more accurately instead of on the surface of the UreG proteins. Also the text for (1) should say “and forms UreE2G2” or “and form the UreE2G2 complex” and for (3) should say urease apoprotein.
299: change “on” to “for”.
307: change to “progress has”.
309: insert “that” after “models”.
318: shift “is” to after “How”.
Many references are lacking italics for microbial names (26, 27, 28, 29, 30, etc.) and have other issues such as the “(80-)” in 15 and 20, “MRNA” and “PH” in 25, subscripts or superscripts in 27, 42 and 67, etc.
Reviewer 2 Report
Title: The Maturation Pathway of Nickel Urease
This review describes the current knowledge on the process of urease maturation by various required accessory proteins. The review is concise and generally well-written. One questions that becomes apparent and is not directly addressed is how UreG bound to Ni and GTP, which dissociates from the UreFD complex, is able to interact with UreFD in the activation complex. Is there a conformational change of UreFD when bound to apo-urease that allows UreG to interact? This could be discussed in the context of models of the activation complex, which are briefly mentioned at the end of the article. Apart from this, there are only a few minor revisions I would like to see.
Minor revisions:
1. Lines 13 - 14: The abstract states that the transfer of Ni proceeds from UreEàUreGàUreF/UreDàurease. This implies that UreFD bind Ni, but I saw no indication of this from the text.
2. Line 27: add the word “the” to read, “the mechanism...”
3. Line 29: the statement “H. pylori causing peptic ulcer and gastric cancer” is misleading. While H. pylori infection is correlated with higher rates of gastric cancer, I do not believe there is evidence of a direct causal link.
4. Line 35: Change “delivery” to “deliver”.
5. Figure 1: The colors of the structure in A do not match the colors of the subunit labels. Also, what is the pink area intended to represent?
6. Figure 3: The PDB ID’s for these structures should be listed in the legend.
7. Line 188: Change “binding” to “bind”
8. Line 236: Change “pre-mature” to “premature”
9. Line 307: Change “progresses have” to “progress has”
